# Theory Content, Question-Behavior Effects, or Form of Delivery Effects for Intention to Become an Organ Donor? Two Randomized Trials

**DOI:** 10.3390/ijerph16071304

**Published:** 2019-04-11

**Authors:** Frank Doyle, Karen Morgan, Mary Mathew, Princy Palatty, Prashanti Kamat, Sally Doherty, Jody Quigley, Josh Henderson, Ronan O’Carroll

**Affiliations:** 1Department of Health Psychology, Royal College of Surgeons in Ireland, D02 DH60 Dublin, Ireland; 2Perdana University-Royal College of Surgeons in Ireland School of Medicine, Selangor, Serdang 43400, Malaysia; karenmorgan@perdanauniversity.edu.my; 3Department of Pathology, Kasturba Medical College, Manipal Academy of Higher Education, Karnataka, Manipal 576104, India; mary.mathew@manipal.edu (M.M.); kamat.prashanti@gmail.com (P.K.); 4Department of Pharmacology, Father Muller Medical College, Mangalore 575002, Karnataka, India; drprincylouispalatty@gmail.com; 5RCSI Bahrain, Kingdom of Bahrain, Adliya, P.O. Box 15503, Bahrain; sdoherty@rcsi-mub.com; 6Department of Psychology, University of Stirling, Stirling FK9 4LA, UK; jody.quigley@postgrad.manchester.ac.uk (J.Q.); jhendy10@hotmail.co.uk (J.H.); ronan.ocarroll@stir.ac.uk (R.O.)

**Keywords:** organ donation, affective attitudes, question-behavior effect, randomized trial, psychological theory, public health

## Abstract

Eliciting different attitudes with survey questionnaires may impact on intention to donate organs. Previous research used varying numbers of questionnaire items, or different modes of intervention delivery, when comparing groups. We aimed to determine whether intention to donate organs differed among groups exposed to different theoretical content, but similar questionnaire length, in different countries. We tested the effect of excluding affective attitudinal items on intention to donate, using constant item numbers in two modes of intervention delivery. Study 1: A multi-country, interviewer-led, cross-sectional randomized trial recruited 1007 participants, who completed questionnaires as per group assignment: including all affective attitude items, affective attitude items replaced, negatively-worded affective attitude items replaced. Study 2 recruited a UK-representative, cross-sectional sample of 616 participants using an online methodology, randomly assigned to the same conditions. Multilevel models assessed effects of group membership on outcomes: intention to donate (primary), taking a donor card, following a web-link (secondary). In study 1, intention to donate did not differ among groups. Study 2 found a small, significantly higher intention to donate in the negatively-worded affective attitudes replaced group. Combining data yielded no group differences. No differences were seen for secondary outcomes. Ancillary analyses suggest significant interviewer effects. Contrary to previous research, theoretical content may be less relevant than number or valence of questionnaire items, or form of intervention delivery, for increasing intention to donate organs.

## 1. Introduction

Organ donation is a successful intervention, with most 5-year survival rates in excess of 70%. However, there is a chronic shortage of organs available for transplantation, suggesting a need to vastly increase the number of either live or posthumous donations. For example, despite the fact that 95% of Americans support organ donation, only 54% have signed up as donors, with currently over 116,000 people on the national transplant waiting list (https://www.organdonor.gov/statistics-stories/statistics.html; [1]). Internationally, the gap between public support and registration can be larger (e.g., http://www.organdonation.nhs.uk; [1]). Therefore, studies investigating how to increase donation rates are of vital international importance.

One particular focus of organ donation research concerns the theoretical content of communication to enhance donation intentions. Recent US and UK findings have shown that cognitive evaluations of the evidence around organ donation are actually less important than emotional or visceral beliefs or reactions, known as ‘affective attitudes’ [1,2,3]. For example, studies have shown that affective attitudes were typically associated with registration behavior, whereas cognitive attitudes were not [1,2]. Examples of negative affective attitudes are the ‘ick factor’—a basic or visceral reaction of disgust to the thought of organ donation; or even the ‘jinx factor’—a fear factor or superstitious belief that simply registering for organ donation will increase the probability that something bad or fatal will happen to an individual. A positive affective attitude would be ‘perceived benefits’, such as bringing meaning to the death of a loved one. These ‘ick’ or ‘jinx’ factors have been associated with lower intention to donate, whereas cognitive attitudes or subjective norms were not [1,2,3]. 

O’Carroll and colleagues have argued that the use of anticipated regret interventions could perhaps overcome these affective attitudes, yielding higher intention to donate and even higher donor registration [3,4,5]. One possible mechanism of action is that anticipated regret may help overcome a “gut feelings”/affective attitudes (system 1 decision making) response, e.g., disgust/“no thanks”. Asking people to think about whether they would regret it later if they did not become a posthumous organ donor encourages reflection and deliberation (system 2 decision making) and could raise intention to become an organ donor [6].

The hypothesis was that as people are motivated to avoid regret, promotion of anticipated regret should motivate people to undertake action to avoid future emotional consequences—i.e., having higher anticipated regret over not donating organs would actually increase rates of donation. For example, self-reported intention to donate was increased by using two items on anticipated regret in a study of 193 non-donor participants [3]. However, these findings failed to replicate in a study of 9,139 nonregistered donors, using verified organ donor registration as the primary outcome [5]. The INORDAR trial randomly assigned participants to one of four groups: a no questionnaire control group, who simply received a letter inviting them to consider organ donor registration; a questionnaire control group, who received a letter and questionnaire with items assessing intention and affective attitudes; a Theory of Planned Behavior (TPB) group, who also received items assessing TPB constructs along with the same items as the questionnaire controls; and an anticipated regret group, who received all items as above, but also two extra items assessing anticipated regret. Six months after receipt of the invitation/questionnaires, the highest rate of registration was actually seen in the no questionnaire control group (6.39% versus 4.51–5.40% in other groups). Thus, this well designed, theory-based intervention actually did harm by *reducing* donor registration [5].

Why did this intervention fail? The authors speculated that this may have been the result of negative priming effects i.e., people were cued to think more negatively about organ donation by completion of negative affective attitude items and this may have counteracted any positive effects of the anticipated regret intervention. This was tested by Doherty et al. [1] in 349 non-donors, using the same questionnaire items as per the INORDAR trial. Doherty et al. [1] tested 3 groups, using an interviewer methodology, with intention to donate as the primary outcome: the first group replicated the anticipated regret group from INORDAR, the second group used the same items, but omitted affective attitudes, while the third group omitted negatively-worded affective attitudinal items only. The second group reported significantly higher intention to donate than the replication group, and were marginally more likely to take an organ donor card from interviewers after the study (an effect entirely mediated by intention), supporting the hypothesis that the assessment of affective attitudes may have counteracted any positive benefits of an anticipated regret intervention.

However, there may be other explanations for these findings. The mere measurement effect, or question-behavior effect, is defined as the influence of any sort of questioning on subsequent performance of any given behavior, and recent meta-analyses have shown a small effect on behavior [7,8,9]. While the INORDAR study used filler/dummy items to ensure the same length of questionnaire across the three intervention groups, the number of questionnaire items in the Doherty et al. [1] study differed per group as dummy items were not utilized. It is therefore possible that simply including more items decreased intention to donate and indeed actual donor registry behavior, not as a result of the actual theoretical content of the questionnaires. Furthermore, the form of delivery differed between the studies. INORDAR used postal methods, whereas Doherty et al. [1] used student interviewers to ascertain people’s intentions to donate. It is therefore possible that a social desirability element played a role in these findings [10,11,12]. Furthermore, the form of delivery may interact with the theoretical content [13]. For example, face-to-face interventions appear to be more effective than online versions of the same intervention [14], while ‘therapist effects’ are well established—where variability in therapists accounts for some of the effects of a given intervention [15,16]. A further potentially important issue is whether different countries and cultures respond differently to these theoretical constructs [17,18], which was not measured in previous studies. We therefore extend previous research here by addressing these issues in two related randomized trials. Specifically, we aimed to determine whether intention to donate organs differed among groups exposed to different theoretical content, but similar questionnaire length, in different countries. We also investigated whether findings were replicated using different forms of delivery. Objectives were as follows:-To replicate the findings of Doherty et al. [1], by determining whether intention to donate was different among groups exposed to differential levels of affective attitudes, when using a similar number of questionnaire items in each exposure-Determine whether findings were similar in different countries and cultures-Ascertain whether results are similar with face-to-face or web-based methodologies

## 2. Materials and Methods 

### 2.1. Context

Ireland, Malaysia and India do not have an organ donor registry, while the UK does. Potential donors in the first three countries can indicate their willingness to donate in nationally-accepted formats, i.e., carrying a signed donor card (or using the approved app; Ireland only), endorsing own driver’s license as a potential future donor, discussing one’s future wishes with family. As next-of-kin can veto organ donation, even with a prior indication of willingness to donate from the potential donor, we primarily used this variable to classify people as non-donors. That is, donors were defined as those who had not discussed their intentions to become donors with family and fulfilled one of the following criteria: carried the donor card, used the donor app or indicated their wishes on their driver’s license [1]. Those who had already donated an organ were classified as donors (n = 13). For data collected in the UK, participants who had not yet signed up to the national registry were defined as non-donors.

Two randomized trial were conducted, using largely the same item content and procedures as per Doherty et al. [1] and the original INORDAR questionnaire items (UK data). Study 1 was pre-registered: NCT02825862. Study 2 was not. We follow the CONSORT and TIDiER statements for appropriate RCT and intervention reporting [19,20] (Appendix A and Appendix B).

### 2.2. Trial Design

In both studies a cross-sectional, three-arm, parallel randomized trial with 1:1:1 allocation ratio design was used. Study 1 was originally to recruit in Bahrain, but this site withdrew and Manipal, India was instead added. No changes to intervention or eligibility criteria were made.

### 2.3. Participants

Participant eligibility criteria were as follows: adults, 18 years and older. Refusal to participate was the only exclusion criteria, except for Malaysia, where non-English speakers were also excluded. Only non-donors (see above) were included in the randomized trial and were asked to provide outcome measures etc. The protocol was approved by the Research Ethics Committees (RECs) in each site (RCSI Dublin REC1048bb; UK—Stirling University General University Ethics Panel [No. 188]; Malaysia, RCSI PU REC—PUIRBH0097; India—Kasturba Hospital Institutional Ethics Committee IEC 134/2017). An opportunistic sample in each country was recruited (study 1). A nationally-representative sample of UK non-donors was recruited in study 2 using an online methodology (Qualtrics subject panel). After provision of a study information leaflet and consenting to participate, participants provided demographic information and then answered 27 questions about organ donation beliefs. 

### 2.4. Interventions and Procedures

For both study 1 and 2, allocated groups were as follows: -Group 1: No-intervention replication group, who completed the entire questionnaire (similar to the anticipated regret group from INORDAR), replicating methods of O’Carroll et al. [3].-Group 2: Omitted affective attitudes group. All 16 questions on affective attitudes were deleted, with dummy questions (e.g., about politics) substituted for these deleted questions (as in the INORDAR trial).-Group 3: Omitted 12 negatively-worded affective attitudes only, which were substituted with the same questions as Group 2.

We tested a question-behavior effect with or without affective attitudes, assessing whether completion of affective attitude items reduces intention to donate, but including dummy items to overcome the mere measurement (question-behavior) effect. Two modalities of intervention delivery were tested to determine if a social desirability effect, i.e., being in the presence of an interviewer, masks the effects of including affective attitudes on intention to donate [21]. 

### 2.5. Materials/Measures 

Three separate questionnaires, one for each group allocation, were used. Materials are available from the authors. All items used 7-point ratings (1 indicating lowest agreement, 7 maximum agreement). We adopted the measures from INORDAR (replicating the procedures used in Doherty et al. [1]), with minor wording changes to reflect the fact that there is no organ donor registry in Ireland, India or Malaysia. In the UK site the original wording was used (both studies). For example, for the primary outcome, intention to donate, non-donor participants were asked the following: -UK: “I will definitely register for organ donation in the next few months”-Non-UK: “I will definitely sign up for organ donation and discuss this with my family in the next few months”

Questionnaires assessed the following theoretical content:

*Affective attitudes* were assessed using 16 items, tapping areas such as the *ick factor* (three items, alpha = 0.83, e.g., “The idea of organ donation is somewhat disgusting”), the *jinx factor* (three items, alpha = 0.72, e.g., “People who donate their organs risk displeasing God or nature”), *medical distrust* (four items, alpha = 0.83, e.g., “If I sign an organ donor card, doctors might not try so hard to save my life”), *perceived benefits* (four items, alpha = 0.61, e.g., “Organ donation allows something positive to come out of a person’s death”), and *bodily integrity* (two items, alpha = 0.78, e.g., “Organ donors may not be resurrected because they don’t have all their ‘parts’”).

*Anticipated regret* was assessed using two items (alpha = 0.79, e.g., “If I did not sign up for organ donation (with card, app or license) and discuss this with my family [or, in the UK: ‘register for organ donation’] in the next few months I would feel regret”).

Theory of planned behavior constructs, *cognitive attitudes* (alpha = 0.86, two items, e.g., “I support the idea of organ donation for transplantation purposes”), *perceived behavioral control* (three items, alpha = 0.68, e.g., “how much control do you have over signing up for organ donation in the next few months?”), and *subjective norms* (two items, alpha = 0.65, e.g., “Most people who are important to me think I should sign up for organ donation and confirm this with my family in the next few months”) were also assessed. *Dummy items* included statements such as “Organ donation is a private matter” or “Lack of organ donation is a serious issue”.

### 2.6. Procedures

In study 1, undergraduate medical students recruited participants in Ireland (in 4 shopping centers in Dublin, dates ranging from 20th January to 19th February 2016 and 12th January to 1st February 2017) and Malaysia (in and around a number of university campuses, dates ranging from July to December 2016). In the UK, a postgraduate student (MSc in Psychology) recruited participants in 11 public libraries in Stirling (12th May to 28th August 2017). In India, the researchers (MM and PP) recruited participants from the University campus of Manipal, Rotary Club Manipal and students from St. Aloysius College, Mangalore from 19th September 2017 to 13th October 2017. No specific standardized training was given, but investigators were available to clarify any issues on request. Study 2 did not utilize interviewers (and was therefore double-blind, whereas study 1 was single-blind). Questionnaires were distributed in different ways. In study 1, the Irish site interviewers used iPads, with paper-and-pencil being used in the other sites. Interviewers approached potential participants, who were interviewed if they consented to this single anonymous interview. Interviewers then completed questionnaires on behalf of participants. After the interview, where applicable (not in the UK), it was also recorded whether interviewers offered an organ donor card and whether participants accepted the card (leaflet in the UK). Apart from the UK, all study 1 interviews were face-to-face, typically lasting 10–20 minutes. In the UK, participants were given questionnaires to complete in the library which they then returned to the researcher. In study 2, the mode of delivery was web-based, recruiting a population-representative sample of UK adults. Participants (aged 18+) from England and Scotland who had never previously donated an organ and were not registered as organ donors were asked to take part in a digital survey (using the U.K. Qualtrics participant panel) in September–October 2017. All participant data were captured digitally through online questionnaires administered by a Qualtrics digital platform. Participants viewed study information and were asked to provide their informed consent to the digital survey. On completion of the survey, participants were thanked and given a debrief statement about the study. Participants were free to leave the survey at any time and also leave questions blank if they wished, or could start and finish the items at any time. Fidelity tests were not conducted.

### 2.7. Outcomes

Intention to donate, assessed by two items on a 7-point scale (e.g., “I will definitely sign up for organ donation and discuss this with my family in the next few months”), was the primary outcome. The secondary outcome (where applicable), was taking a donor card after the interview (study 1; yes/no), or asking participants if they would like to be taken to the organ donor registry website (study 2; yes/no).

### 2.8. Sample Size

As recruitment was expected to vary considerably across sites, a formal power calculation was not conducted, but instead we aimed to recruit as many participants as possible within the allocated timeframes. We estimated a possible 750 participants for study 1 based on Doherty et al. [1], and a lower number for study 2. This would also allow for within-site differences to be explored. 

### 2.9. Randomization

Study 1: Participants were block-randomized (random block sizes ranging from 3–15) using the *ralloc* command for Stata 13.0 (StataCorp LLC, College Station, TX, USA). FD supplied eight separate (four for Dublin, two for Malaysia, two for India) participant recruitment Excel sheets to the Dublin interviewers and the lead investigators in Malaysia and India, which showed the order in which participants should be recruited. In the UK (study 1), participants were randomly allocated using a random number generator (researchrandomiser.org). Participants were then recruited sequentially, as per the random recruitment order. Therefore, the participants were blinded to group allocation, but interviewers were not. In Study 2 the Qualtrics software automatically randomized the on-line participants to one of the 3 conditions using simple randomization, and this was therefore double-blinded.

### 2.10. Statistical Methods

As per CONSORT guidelines, we did not ascertain between group differences at baseline [19,22]. As this is a cross-sectional study, with outcomes recorded at the same time as exposures, imputation procedures were not considered, and all participants were analyzed as per original group allocation. Using group 1 (INORDAR replication group) as the reference, we used multilevel linear and logistic modelling to ascertain differences in intention score and taking of organ donor cards among groups within each site, with countries as the random intercept, using the *mixed* and *melogit* commands in Stata 15.0 (StataCorp LLC, College Station, TX, USA). We ran three main models, for study 1, study 2 and then combined. We also ran sensitivity analyses, omitting the UK face-to-face data, as this involved relatively little interaction with the researcher. In *post-hoc* analysis, we explored the Irish data only, as this dataset contained variables indicative of more levels within the data (4 different interviewers). 

## 3. Results

### 3.1. Participant Profile

The participant flow chart is shown in Figure 1. 

Data on non-responders were not collected. In study 1, a total of 1007 non-donors participated, of whom 341, 323 and 343 were allocated to groups 1–3 respectively, with outcomes ascertained for 312, 281, 309 (intentions) and 300, 289 and 306 (taking donor card) respectively. In study 2, 612 non-donors were assigned to groups 1–3 in the following numbers: 209, 208 and 195, with intention scores and the secondary outcome of linking to the organ donor website available for all.

Baseline data is available in Table 1, stratified by groupMean age for study 1 was lower than for study 2 and approximately half of the sample in each study were women. Overall, approximately one third had donated blood. In the combined data, 15.5% knew someone who had received an organ, 8.76% knew someone who needed an organ and 10.7% knew someone who had donated an organ, although prevalence varied among studies and conditions. Variables differed considerably across countries (Appendix C).

### 3.2. Outcomes

Outcomes are summarized in Table 2. Mean intention scores were overall lower for study 2 than for study 1, and proportions of people who transferred to the organ donor website were much lower than the proportions who accepted a donor card. 

Table 3 shows the results of the multi-level models predicting intention and behavior by group allocation. There were no significant differences seen for primary or secondary outcomes among groups for either study 1 or the combined data. In study 2, group 3 demonstrated a small but statistically higher intention score in comparison to group 1, but this did not translate into behavior, with no differences among groups for subsequent linkage to the UK organ donor website. A sensitivity analysis, omitting the UK face-to-face data, did not change the results (data not shown).

### 3.3. Ancillary Analyses

In exploratory analyses, we also estimated the effects of different researchers on outcomes in the Irish data only, as this was the only dataset to record the interviewers. There was significant impact on intention scores, irrespective of group, depending on the researcher involved, with no impact by group exposure (Table 3).

## 4. Discussion

In two randomized studies, the overall results suggest no effects of affective attitudinal content on intention to donate or subsequent donor behaviors when the number of questionnaire items is held constant. In study 1, a single-blind, multi-country, face-to-face delivered intervention, no differences in primary or secondary outcomes were seen according to affective attitudinal content. In study 2, which was double-blind and web-based, a small, but significantly higher intention score was seen for the group which omitted negatively-worded affective attitudinal items, but this did not translate into increased organ donor behavior via linkage to the organ donor registration website.

The results are in direct contrast to previous research and fail to replicate the findings of Doherty et al. [1], suggesting that when the numbers of questionnaire items are held constant, there was no effect of theory-content on intention to donate or donor behavior. It could be that the question-behavior effect in this instance is powerful enough to overcome the negative contextual cueing priming effect of including affective attitudes [5]. The results also suggest a potential reason for the failure of the INORDAR trial [5]. One reason for the lower donation rates in this trial may simply be due to having a higher number of questionnaire items in the intervention groups, when compared to a no-questionnaire control group. This could also suggest that theoretical content was simply less important than number of questionnaire items participants had to complete. 

The differences in results between study 1 and 2 deserve comment. First, intention scores and behaviors were significantly lower in the web-based study 2 than in the interviewer-based study 1, supporting the notion that having interviewers can lead to social desirability bias in respondents’ answers and behaviors [8,10,11,12]. Social desirability has been shown to be related to higher intention to donate scores, albeit not higher proxy donation behaviors, in a recent study on reciprocal altruism in which the face-to-face interviews were conducted by a single researcher [21]. This is similar to the findings of study 2—where the negatively-worded affective attitudes omitted group showed highest intention to donate. Therefore, the social desirability elements of study 1 may have masked these effects. The major caveat is, however, that there was no effect on behavior, even with higher intention scores, as has been observed in previous research [21]. Second, an alternative possible explanation for differences in findings from study 1 to study 2 is the modification of questionnaire items needed for countries without an organ donor register. However, this is unlikely to be the case, as the UK data from study 1 also had non-significant results. Therefore, the differences in intention scores and behaviors between studies 1 and 2 are more likely to be due to the different forms of delivery [13]. Third, although the study was conducted across countries with very different cultural contexts, the results were similar in that no effects for group membership were seen in individual countries (data not shown), suggesting a robust effect of questionnaire length on outcomes. Future research could consider countries with high organ donation rates (e.g., Spain) or legislative frameworks (e.g., opt-in versus opt-out), but also the target populations (living donor versus post-mortem donors), to determine if these influence attitudes and overall results. 

As mentioned above, therapist-specific effects are well-established in other areas of research [15,16], including charitable donation [23]. Indeed, recent research in smoking cessation intervention suggests that individual counselors explain up to 9% of the variance in cessation outcomes, and cessation counselors with higher extraversion scores may be more effective [24,25]. The ancillary analyses from the present study support this finding in terms of organ donation, with significantly different primary and secondary outcomes reported in the Irish data, which vary not as a result of group membership, but by interviewer. This replicates and supports the findings of Doherty et al. [1], who also demonstrated significantly different intention and behavior scores by interviewers. While it is in contrast to previous research which has suggested that the form of delivery of organ donation communications is not relevant [26], the current findings, along with those of Doherty et al. [1], suggest that the individual who is delivering the intervention may be a critical determinant of organ donation research in particular, health psychology and behavioral medicine research more generally. Form of delivery as an ‘active ingredient’ of behaviour change interventions has largely been ignored to date [13].

### Limitations and Strengths

We acknowledge several limitations inherent in the study. Self-reported intention was the primary outcome, albeit we had (proxy) behavioral measures as secondary outcomes. The design is cross-sectional, so we have no indication of subsequent behavior or attitudes to organ donation, or indeed any ability to account for initial elevation bias in the questionnaires [27]. Minor wording changes were required in study 1 for 3 of the participating countries, where discussions with family and becoming a donor were possibly confounded in the items, although this was not needed for study 2. The design of study 1 was single-blind and it is possible that interviewers approached participants with reference to the group that was next on the randomization list—but, again, this was not the case in study 2. However, despite these limitations, results were consistent across both outcomes, suggesting a robust effect. Anticipated regret was included in all groups and it is possible that results may be different if anticipated regret items were not included in one or more groups. A recent meta-analysis has shown that affective forecasting has a small initial effect on anticipated regret, but effects on later intentions or regret were not significant [28]. Future research should address this, by including a trial arm that does not contain anticipated regret. A power calculation was not conducted beforehand, as we simply required students to recruit for the duration of their research placement (study 1), and we knew we would obtain sufficient numbers with four sites recruiting. The sampling techniques used in study 1 limit the generalizability of the results, however, in contrast, we did recruit from four countries and study 2 does involve a population-generalizable sample. We also used 2 methodologies—face-to-face and web-based, increasing generalizability. Other strengths included adopting already-used methods and items from previous research. Another limitation is that we did not adopt a no-questionnaire control group, to compare organ donor intentions and behaviours without exposure to questionnaire (theory) content, as in the original INORDAR trial. Future work should address this. 

## 5. Conclusions

In two large randomized trials, intention to become an organ donor was not influenced by the theoretical content of the questionnaire when similar numbers of items were used across conditions, and was only influenced by theoretical content when the form of delivery was internet-based. When testing theory, researchers must use similar numbers of items across conditions to guard against the mere measurement effect, and fully report the form of intervention delivery. There may also be important interventionist effects to consider and investigate in future research.

## Figures and Tables

**Figure 1 ijerph-16-01304-f001:**
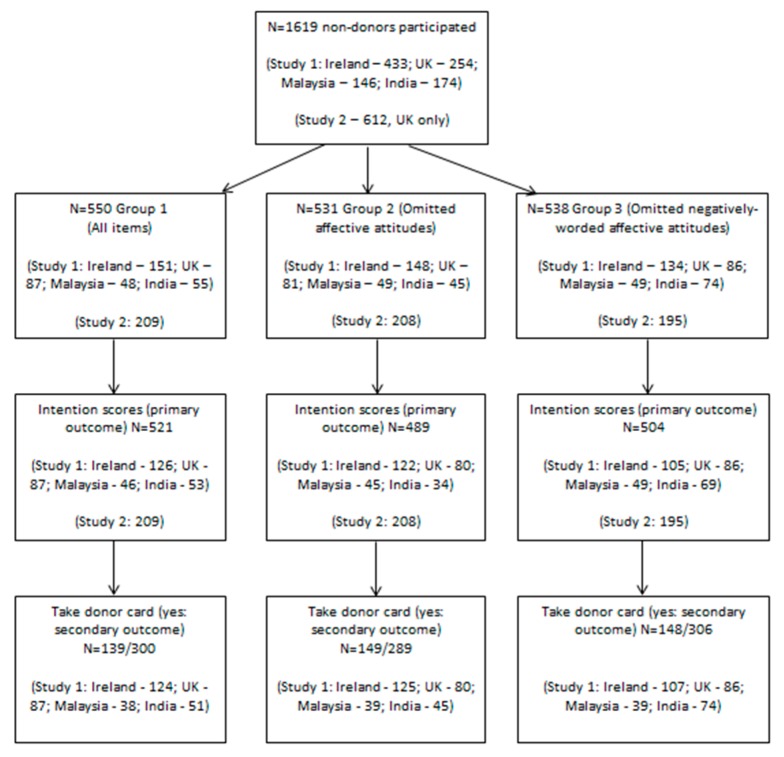
Participant flowchart and outcome ascertainment.

**Table 1 ijerph-16-01304-t001:** Sample description.

	Study 1					Study 2				Combined Data			
	Study 1, N	Study 1, All Groups	Group 1	Group 2	Group 3	Study 2 Overall (N = 612 for All)	Group 1	Group 2	Group 3	Combined Samples	Group 1	Group 2	Group 3
Age (mean, SD)	1005	34.4 (18.1)	34.6 (17.9)	34.8 (18.1)	33.7 (18.2)	50.4 (15.6)	50.8 (15.3)	49.5 (16.1)	51.0 (15.4)	40.4 (18.9)	40.7 (18.7)	40.6 (18.8)	40.0 (19.1)
Women	1004	54.0%	50.7%	56.1%	55.3%	50.3%	51.2%	52.4%	47.2%	52.6%	50.9%	54.6%	52.3%
Private health insurance †	691	50.7%	51.2%	47.2%	53.7%	†	†	†	†	-	-	-	-
Interview location (anonymized, Ireland only)	433					n/a	n/a	n/a	n/a	-	-	-	-
1		26.3%	30.5%	22.3%	26.1%								
2		34.2%	33.8%	35.2%	33.6%								
3		27.5%	23.2%	32.4%	26.9%								
4		8.08%	7.95%	6.76%	9.70%								
5		3.93%	4.64%	3.38%	3.73%								
Do you know someone who (% yes)													
has received an organ	992	18.2%	17.0%	19.0%	18.5%	11.1%	12.0%	10.6%	10.8%	15.5%	15.1%	15.7%	15.7%
needs an organ	997	11.9%	10.0%	13.6%	12.3%	3.59%	4.78%	4.33%	1.54%	8.76%	8.01%	9.92%	8.40%
has donated an organ	994	13.1%	11.9%	15.1%	12.3%	6.86%	8.61%	5.29%	6.67%	10.7%	10.6%	11.2%	10.3%
Ever donated blood	998	34.6%	33.1%	37.3%	33.4%	31.2%	37.3%	28.9%	27.2%	33.3%	34.7%	34.0%	31.2%

† National health insurance for all citizens in UK; n/a—not applicable.

**Table 2 ijerph-16-01304-t002:** Summary of outcomes (intention and taking card/website transfer).

	Study 1					Study 2				Combined Data			
	Study 1, N	Study 1, All Groups	Group 1	Group 2	Group 3	Study 2 Overall (N = 612 for All)	Group 1	Group 2	Group 3	Combined Samples	Group 1	Group 2	Group 3
Intention (mean, SD)	902	4.41 (1.60)	4.44 (1.66)	4.30 (1.57)	4.47 (1.56)	3.25 (1.54)	3.13 (1.49)	3.19 (1.56)	3.44 (1.56)	3.94 (1.68)	3.91 (1.72)	3.83 (1.66)	4.07 (1.64)
Taking donor card (yes, %)	895	48.7%	46.3%	51.6%	48.4%	-	-	-	-	-	-	-	-
Transferred to organ donor website	-	-	-	-	-	7.03%	6.22%	6.25%	8.72%	-	-	-	-
Combined behavioral outcome (Taking card or website transfer)	-	-	-	-	-	-	-	-	-	31.8%	29.9%	32.6%	32.9%

**Table 3 ijerph-16-01304-t003:** Results of regression models predicting outcomes.

		Primary Outcome (Intention)				Secondary Outcomes (Card/Website)		
	Intention	β	Std. Err.	95% CI	*p*	OR	95% CI	*p*
Study 1	Group 1	Ref.	-	-	-	Ref.	-	-
	Group 2	−0.120	0.130	−0.375 to 0.134	0.354	1.28	0.890 to 1.84	0.183
	Group 3	0.025	0.127	−0.224 to 0.274	0.850	1.34	0.932 to 1.93	0.113
Study 2	Group 1	Ref.	-	-	-	Ref.	-	-
	Group 2	0.061	0.150	−0.234 to 0.355	0.686	1.00	0.426 to 2.03	0.990
	Group 3	0.307	0.153	0.007 to 0.606	0.044*	1.44	0.680 to 3.05	0.341
Combined data	Group 1	Ref.	-	-	-		-	-
	Group 2	−0.047	0.098	−0.239 to 0.146	0.637	1.22	0.882 to 1.70	0.227
	Group 3	0.136	0.098	−0.055 to 0.328	0.163	1.37	0.982 to 1.90	0.063
Exploratory analyses—Irish data only	Group 1	Ref.	-	-	-	Ref.	-	-
	Group 2	−0.124	0.199	−0.514 to 0.266	0.533	1.55	0.916 to 2.63	0.102
	Group 3	−0.184	0.207	−0.590 to 0.221	0.373	1.52	0.877 to 2.63	0.136
	Researcher 1	Ref.	-	-	-	Ref.	-	-
	Researcher 2	−0.809	0.250	−1.30 to −0.319	0.001 **	0.42	0.210 to 0.851	0.016 *
	Researcher 3	−0.395	0.227	−0.840 to 0.049	0.081	0.34	0.184 to 0.647	0.001 **
	Researcher 4	0.283	0.235	−0.177 to 0.745	0.227	0.63	0.326 to 1.22	0.174

* *p* < 0.05, ** *p* < 0.01; Group 1—replicating INORDAR; Group 2—omitting all affective attitudinal items; Group 3—omitting negatively-worded affective attitudinal items.

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
