# Peer review of "Theory Content, Question-Behavior Effects, or Form of Delivery Effects for Intention to Become an Organ Donor? Two Randomized Trials"

_ijerph, 2019, doi:10.3390/ijerph16071304_

Round 1
Reviewer 1 Report
It would be interesting to do a similar studies with other countries. Countries which has got a high rate of organ donation per million population, like Spain.
Previously, some articles with this INORDAR model has already been published, but I believe these results are original.
Author Response
Yes, these results are original – as stated in the manuscript this paper tries to determine why INORDAR failed to enhance organ donation rates (and indeed caused harm by reducing donation rates in some groups).
We agree that it would be interesting to do similar studies in other countries, which have higher or lower rates of organ donation. We have added this suggestion to the manuscript.
“Future research could consider countries with high organ donation rates (e.g. Spain) or legislative frameworks (e.g. opt-in versus opt-out), but also the target populations (living donor versus post-mortem donors), to determine if these influence attitudes and overall results.”
Reviewer 2 Report
I am afraid that my comments to you as authors are going to seem rather obtuse. This is a well-written paper and the study (to me) seems well-designed and executed with an admirable sample size. I was attracted to review your paper because of what I thought was the substantive topic (intention to become a donor). However your paper is really about survey design and delivery. I note that you do address some of the ethical issues around donor recruitment in your introduction but your paper is not really about those issues and nor do you draw an conclusions about the role of surveys in influencing donor intention.
I find it odd that you don't distinguish between living donors and post-mortem donors (aren't these very different populations?) I also find it odd that you don't mention the increasingly popular move towards opt-out systems for post-mortem donation, about to become law in England, now established in Wales and in many other jurisdiction.
I am afraid that I may have missed the point of your study.
p 5 line 204 there is a missing number.
The ordering of the appendices is odd (A, B, 3) also I do not see the point of appendices A and B.
Author Response
Thank you for your positive comments about design and execution.
We do conclude that surveys that test theory need to use similar numbers of items (see Conclusions).
The reviewer is correct that post-mortem donors and living donors are different populations. However, the original INORDAR trial did not differentiate between these, so therefore we do not in the current manuscript. We have now, however, added this point to the manuscript as an avenue for future research (as per reply to reviewer 1 also):
“Future research could consider countries with high organ donation rates (e.g. Spain) or legislative frameworks (e.g. opt-in versus opt-out), but also the target populations (living donor versus post-mortem donors), to determine if these influence attitudes and overall results.”
We are unsure which number is missing? The dates of recruitment for India were provided on line 205. Numbers recruited were provided in the Results section.
We have changed the labelling of Appendices (3 replaced by C).
Appendices A and B are requirements for submission of complex interventions. We have referred to these in the Methods.
“We follow the CONSORT and TIDiER statements for appropriate RCT and intervention reporting [19, 20] (Appendices A and B).”
Reviewer 3 Report
This article reports on research which examines the impact of quesitonnaire structure and valence on intention to donate organs. The article is well-organised and contains all the components that I would expect: abstract, introduction, materials and methods, results, discussion and conclusions. The sections are generally well-developed although I am suggesting a few minor changes as detailed below to improve the clarity of presentation in some areas.
I found the abstract a little unclear on the first read and suggest that this could be clearer with the insertion of the word 'items' on lines 24-25, to read 'affective attitude items replaced' and 'negatively-worded affective attitude items replaced'.
Lines 110-112 offer a clear statement of the aims of the research, which I feel would have been useful to have included within the abstract.
Section 2.6 Procedures: clear information is provided about the recruitment procedures in Dublin, Malaysia and Stirling; however, recruitment in Manipal is described as being 'from the general public' but does not state how participants were recruited in Manipal.
Line 193: typo - "Lack or organ donation is a serious issue" should read lack of
Line 204: study number is missing
Line 259: 'data ' should read data were
Author Response
Thank you - we have made this change to the abstract.
We have added the recommended sentence to the abstract
Regarding recruitment in India, we have added the following:
"In India, the researchers (MM and PP) recruited participants from the University campus of Manipal, Rotary Club Manipal and students from St. Aloysius College, Mangalore from 19th September 2017 to 13th October 2017."
We have addressed all of the listed typos.
Round 2
Reviewer 2 Report
The missing Number at line 204 (original MS) is I think '1' in reference to study 1 - it reads:
'whereas study was single-blind.'